# TCIG: Two-Stage Controlled Image Generation with Quality Enhancement through Diffusion

## Abstract

In recent years, significant progress has been made in the development of text-to-image generation models. However, these models still face limitations when it comes to achieving full controllability during the generation process. Often, specific training or the use of limited models is required, and even then, they have certain restrictions. To address these challenges, A two-stage method that effectively combines controllability and high quality in the generation of images is proposed. This approach leverages the expertise of pre-trained models to achieve precise control over the generated images, while also harnessing the power of diffusion models to achieve state-of-the-art quality. By separating controllability from high quality, This method achieves outstanding results. It is compatible with both latent and image space diffusion models, ensuring versatility and flexibility. Moreover, This approach consistently produces comparable outcomes to the current state-of-the-art methods in the field. Overall, This proposed method represents a significant advancement in text-to-image generation, enabling improved controllability without compromising on the quality of the generated images.

## 1 Introduction

The current state-of-the-art text-to-image diffusion models (Croitoru et al. (2023), Rombach et al. (2022), Ramesh et al. (2022), Saharia et al. (2022), Yu et al. (2022)) have demonstrated remarkable power in generating a vast array of possible images from a single prompt. However, one limitation of these models is their lack of full controllability. When a user has a rough idea of how the generated image should look, it becomes challenging to convey those preferences through text alone. Specific details such as the exact location and regions of objects often require additional means of control. While some methods have been developed to address this controllability issue, they either involve training the model specifically for this purpose or fine-tuning it (Avrahami et al. (2023), Nichol et al. (2022), Ramesh et al. (2022), Rombach et al. (2022), Wang et al. (2022)). These approaches are constrained by computational power and the availability of training data. Another class of methods achieves controllability without requiring special training (Bar-Tal et al. (2023)), but they also have their limitations.

In this work, A novel method is proposed for generating controlled images that does not necessitate training. The proposed approach involves dividing the generation process into two stages. In the first stage, A pre-trained segmentation model is utilized to generate a highly controlled image based on a reference input segmentation mask. Although this first stage excels in control, it may lack quality and fine details. To address this, the output of the first stage is fed into a diffusion model in the second stage, which produces the final controlled output. By combining the power of a pre-trained segmentation model and a diffusion text-to-image model, TCIG (**T**wo-stage **C**ontrolled **I**mage Generation) enables the generation of controlled images from both text and segmentation mask inputs. Notably, TCIG achieves results comparable to state-of-the-art models, surpassing previous solutions in terms of controllability and overall performance.

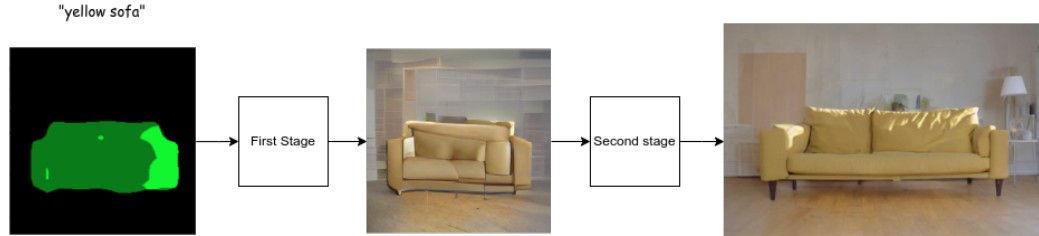

Figure 1: Two stage generation, the first stage generates a perfectly contorlled image and second stage for producing high quality final output.

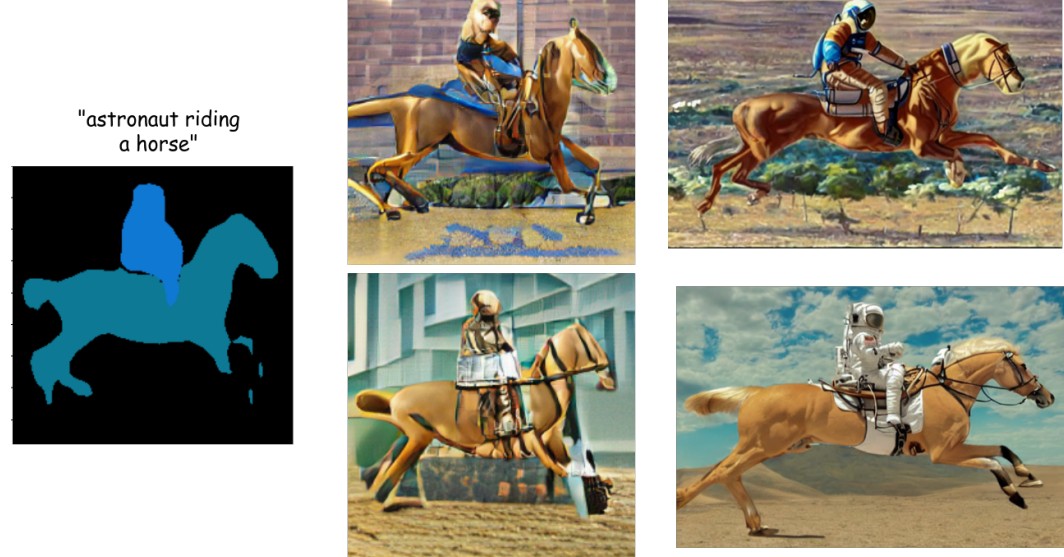

Figure 2: Two stage generation, the first column is the input (text prompt + segmentation masks), second column is two sample outputs of the first stage, and third column is two outputs of the second stage from the two inputs of the second stage respectively.

## 2 RELATED WORK

Diffusion models (Dhariwal & Nichol (2021), Ho et al. (2020), Ramesh et al. (2022), Saharia et al. (2022) ) have emerged as dominant players in various vision tasks, with Latent Diffusion models (LDMs) (Rombach et al. (2022)) in particular revolutionizing the field by performing the diffusion process on a lower-dimensional level. This approach not only saves computational resources but also achieves state-of-the-art results.

In the pursuit of controllability in image generation, several models have introduced an additional form of input in the form of labeled semantic layouts (Ho et al. (2020), Nichol et al. (2022), Ramesh et al. (2022), Rombach et al. (2022), Saharia et al. (2022), Wang et al. (2022)). While these approaches provide some level of control, they often fall short of offering complete control over the generation process. Furthermore, they typically require costly training procedures and specific datasets. Alternatively, some models resort to fine-tuning pretrained models (Kawar et al. (2023), Kim et al. (2022), Meng et al. (2022)), while others manipulate the generation process of a pretrained model (Bar-Tal et al. (2023), , Choi et al. (2021), Couairon et al. (2022), Hertz et al. (2022), Kong et al. (2022) Kwon & Ye (2023), Meng et al. (2022), Mokady et al. (2022), Tumanyan et al. (2022) ). However, these methods heavily rely on the underlying architectural structure and internal details of the pretrained model, making them less versatile and requiring training or fine-tuning.

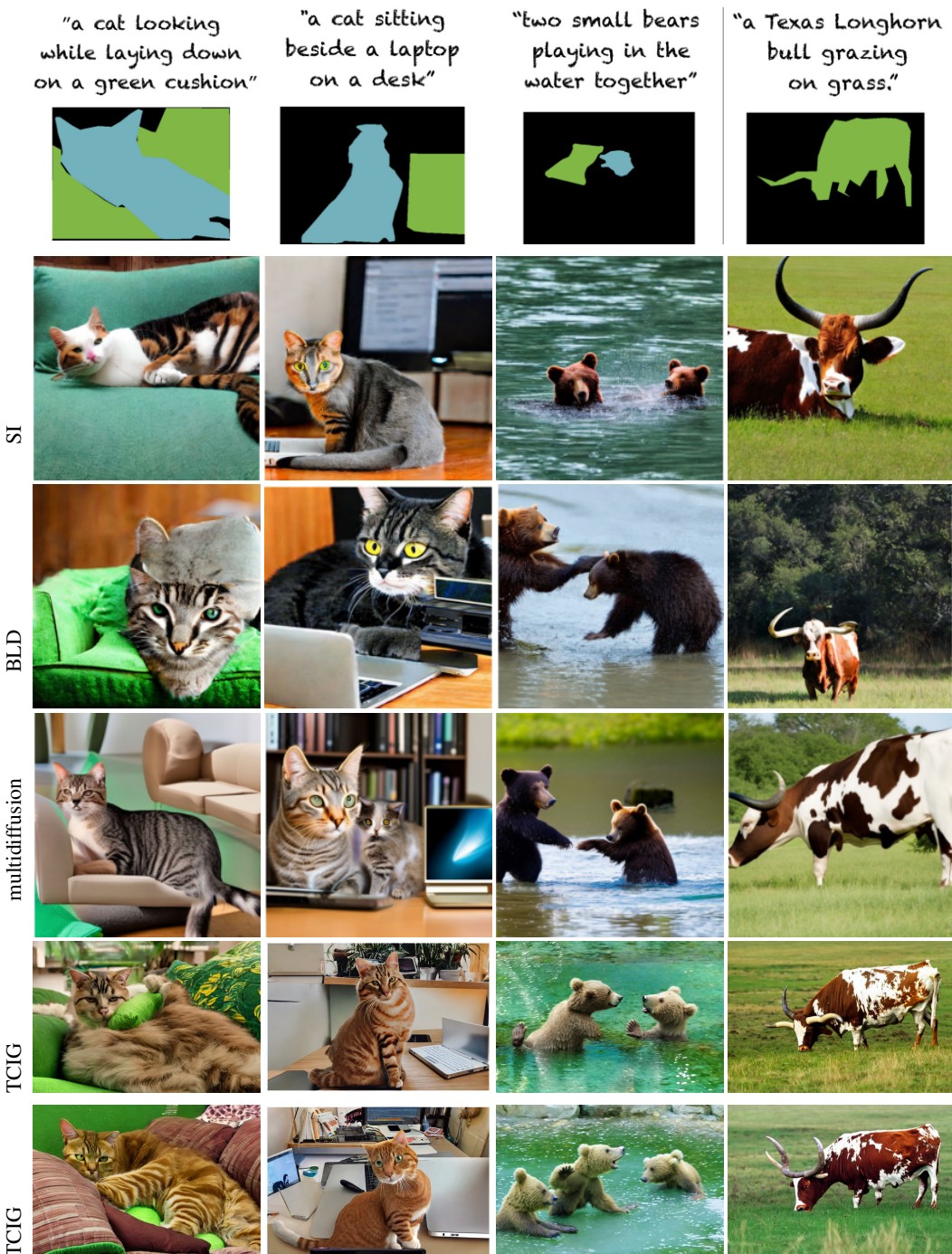

Figure 3: A comparison between this method (TCIG) and others (Avrahami et al. (2022), Bar-Tal et al. (2023), Rombach et al. (2022)). First row contains the input segmentation maps and it's text description prompt, last two rows are two samples of TCIG. figure was adapted from Bar-Tal et al. (2023).

In contrast to existing methods, this work takes a different approach that does not rely on the specific architecture or internal details of the pretrained model, eliminating the need for training or fine-tuning. Instead, A novel method is proposed that harnesses the power of a pre-trained segmentation model and a diffusion text-to-image model to achieve controllability in image generation. By dividing the generation process into two stages. This two-stage approach combines the strengths of both models, providing a powerful and controllable image generation method that rivals state-of-the-art models in terms of performance.

By avoiding the constraints of architecture dependency and costly training procedures, this method opens up new possibilities for generating controlled images without sacrificing quality or controllability.

## 3 METHOD

The model consists of two distinct stages. In the first stage, the objective is to generate an image that closely aligns with the input sketch (segmentation masks) and text, prioritizing the overall resemblance rather than focusing on the specific size and quality of the output image. The second stage, on the other hand, focuses on enhancing the image by increasing its resolution and refining the details, thereby improving its overall quality, see Figure 1 and Figure 2.

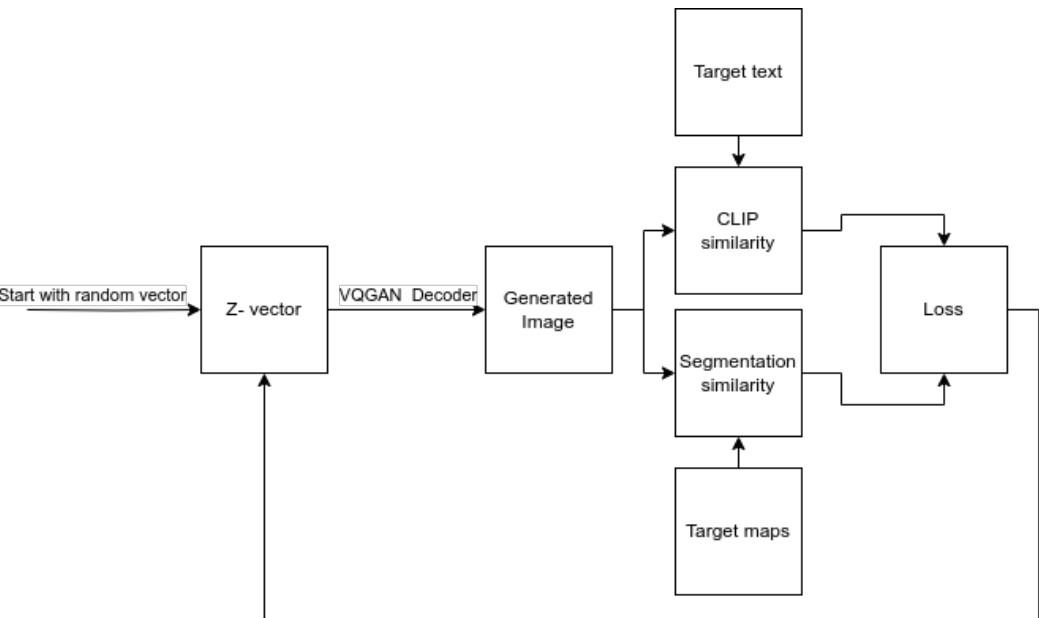

Figure 4: First stage, through the guidance of segmentation models and the CLIP network (Radford et al. (2021)), a controlled image is produced

### 3.1 FIRST STAGE

The VQGAN+CLIP model (Crowson et al. (2022)) enables the generation of images from text prompts by leveraging a pretrained VQGAN. This process involves multiple forward and backpropagation iterations, guided by a CLIP network (Radford et al. (2021)), until it converges to a solution. To enhance this method, expert pretrained segmentation models are utilized to provide additional guidance. This aids in steering the embedding vector of the VQGAN towards an image that closely matches the target masks and achieving a higher level of control as in Figure 4.

$$\mathcal{L}_{seg} = MSE(m_p, m_t) \qquad (1)$$

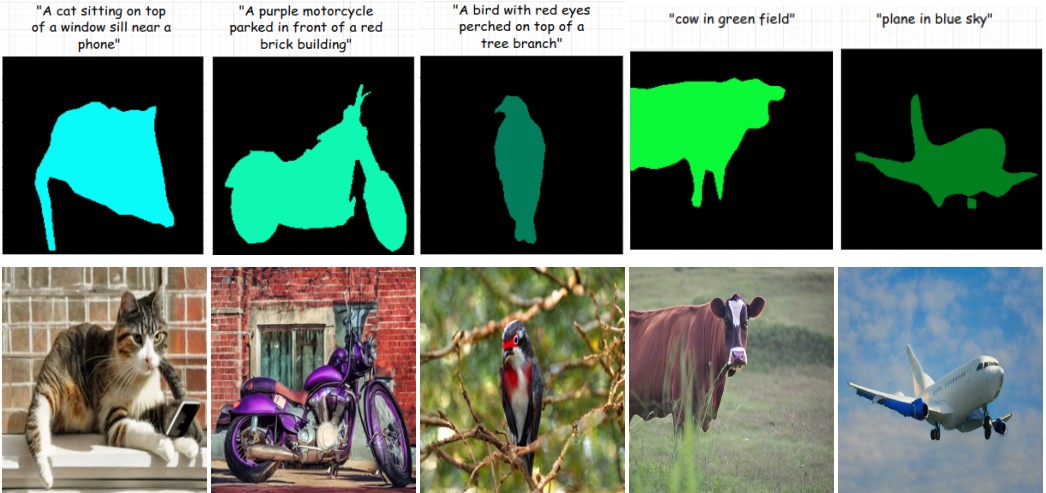

Figure 5: sample outputs of TCIG, First row contains the input segmentation maps and it's text description prompt, second row contains the final output

Equation 1 is the loss calculated for the segmentation model guidance ($\mathcal{L}$seg) where MSE is the mean-squared-error loss function, $m_p$ are the predicted masks and $m_t$ are the target masks. Moreover, the aforementioned approach can be extended to incorporate multiple guiding segmentation models, each specializing in a specific subset of classes. By implementing a straightforward logic based on the input classes, the appropriate guiding models can be selected. This effectively resolves the challenge of employing a cumbersome mask encoder (Gafni et al. (2022)) or expert model, offering a scalable solution that supports unlimited classes. Additionally, to consolidate all components, the loss from the CLIP network (Radford et al. (2021)) is combined with the other losses Equation 2.

$$\mathcal{L}_t = \alpha_c \mathcal{L}_{clip} + \sum_{i=1}^{n} \alpha_{s_i} \mathcal{L}_{seg_i} \qquad (2)$$

Where $\mathcal{L}_t$ is the total loss, $\alpha_c$ is the loss factor for the loss from clip $\mathcal{L}_{clip}$, $\alpha_{s_i}$ is the factor for the loss from segmentation model i.

## 3.2 SECOND STAGE

The second stage of this image generation pipeline uses a pre-trained diffusion model to improve the quality of the generated image, specifically an Img-to-Img pipeline. This is possible because the image is not pure noise, which allows us to separate the controlling process from the refinement process. This is beneficial because it allows us to use state-of-the-art diffusion models to achieve high quality, more details, and a more realistic look, while still maintaining control over the image generation process. Another advantage of this approach is that it is very flexible, as any diffusion method of this type can be used. Additionally, fewer diffusion steps is needed than when generating an image from pure noise. To address the problem of imperfect masks, the parameters are adjusted of the diffusion model to allow for more flexibility. This is necessary because the output from the first stage can overfit the input masks, which are often generated by non-expert users and therefore imperfect.

## 4 EXPERIMENTS

This method allows users to flexibly and controllably generate images (see Figure 5). It generates highly diverse samples because starting with a random vector Z in the first stage converges to a

Table 1: the IoU metric comparison done on the COCO dataset (Lin et al. (2015)) for this method and Avrahami et al. (2022), Rombach et al. (2022) and Bar-Tal et al. (2023). Table was adapted from Bar-Tal et al. (2023)

|  | IoU |
| --- | --- |
| SI | $0.16 \pm 0.10$ |
| BLD | $0.17 \pm 0.11$ |
| multidiffusion | $0.26 \pm 0.12$ |
| TCIG | $\mathbf{0.30 \pm 0.26}$ |

different output, and every first stage output can generate multiple second stage outputs (see Figure 2). Additionally, the generated images comply with both the text and input masks.

First, this method is quantitatively compared with (Avrahami et al. (2022), Bar-Tal et al. (2023), and Rombach et al. (2022)) (see Table 1). To do this, the COCO dataset (Lin et al. (2015)) is used, specifically the validation portion of the data. the DeepLabv3 model (Chen et al. (2017)) is used as the guiding segmentation model and the stable diffusion model (Rombach et al. (2022)) Img-to-Img pipeline. Pascal VOC (Everingham et al. (2010)) classes were only considerd, and like multidiffusion (Ramesh et al. (2022)), the final filtered set consists of images with 2 to 4 foreground objects excluding people and masks that occupy less than 5% of the image. the Intersection over Union (IoU) metric is also used with respect to the ground-truth segmentation (Table 1).

Second, the method is compared qualitatively with (Avrahami et al. (2022), Bar-Tal et al. (2023), Rombach et al. (2022)) noting that not all of these models are public, and it's noticeable that TCIG produce better fitting images with the input masks (see Figure 3).

## 5 CONCLUSION

Controllable image generation has been one of the major challenges in AI, and it remains so today. Despite the computational power limitations of GPUs faced in developing this method, it has managed to achieve a unique methodology for generating controlled images. This method has the advantages of being very flexible and easily integrated with state-of-the-art diffusion models for text-to-image generation. Additionally, it provides more control over the generation process while still allowing for some flexibility to overcome imperfect masks. TCIG's results are comparable to state-of-the-art models, if not setting new state-of-the-art results. This work will trigger further research into separating control from high-quality image generation.

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
