# OpenReview forum: "TCIG: Two-Stage Controlled Image Generation with Quality Enhancement through Diffusion"
_ICLR.cc/2024/Conference — Submitted to ICLR 2024_

### Official Review · Reviewer_nTLn · 2023-10-18

**Soundness:** 1 poor
**Presentation:** 1 poor
**Contribution:** 1 poor
**Rating:** 1
**Confidence:** 5

**Summary:**

This paper leverages VQGAN to generate an initial image based on text and segment map guidance, followed by refinement using a diffusion model. The authors claim that this  algorithm enhances controllability while upholding image quality.

**Strengths:**

The proposed algorithm suggests the potential of harnessing multiple pre-trained models to achieve superior generation results.

**Weaknesses:**

This paper is not yet publication-ready. The proposed algorithm appears to be a fusion of two pre-trained models, lacking a demonstration of its non-triviality. Furthermore, the experiments fail to establish its superiority over existing competitors.

**Questions:**

1. Why is the inclusion of VQGAN necessary in the pipeline? Could the classifier guidance be directly applied to the diffusion model without VQGAN?

2. How to select the coefficients in Eq. (2)?

3. What accounts for the notably larger variance observed in Table 1 when using the proposed method?

4. The paper would benefit from additional details, including the time and memory requirements for generation and an analysis of each component's contribution through ablation studies.

---

### Official Review · Reviewer_UDgM · 2023-10-22

**Soundness:** 3 good
**Presentation:** 2 fair
**Contribution:** 2 fair
**Rating:** 3
**Confidence:** 4

**Summary:**

This paper introduces a two-step method for image generation.

First, it uses a trained model to create a controlled image.

Next, a diffusion model gives the final image.

The method is simple but makes sense.

**Strengths:**

The method seems to work.

The paper is easy to understand.

The steps are clear.

**Weaknesses:**

The method lacks of novelty. Seriously.

It misses some important related works, e.g., SceneComposer: Any-Level Semantic Image Synthesis, CVPR 2023.

Figures, like Fig. 2, need more details.

More example images are needed.

**Questions:**

Please add FID or CLIP scores for comparison.

The paper needs more work before it's ready.

Better figures and more examples will help.

**Details Of Ethics Concerns:**

Please review and mention more related works.

---

### Official Review · Reviewer_pNhh · 2023-11-01

**Soundness:** 1 poor
**Presentation:** 1 poor
**Contribution:** 1 poor
**Rating:** 1
**Confidence:** 5

**Summary:**

This paper proposes a two-stage framework to generate controlled images. Specially, the first stage generates a controlled image and second stage for producing final output.

**Strengths:**

None

**Weaknesses:**

This paper is too rough and does not meet the standards of top-tier conferences.

**Questions:**

This paper is too rough and does not meet the standards of top-tier conferences.

---

### Official Review · Reviewer_anks · 2023-11-01

**Soundness:** 1 poor
**Presentation:** 1 poor
**Contribution:** 1 poor
**Rating:** 1
**Confidence:** 5

**Summary:**

This paper proposes a two-stage method to combine controllability and high quality in image generation. In the first stage, the authors utilize pre-trained VQGAN and segmentation models for precise layout control. In the second stage, they feed the generated image to a  diffusion model for a enhanced high-quality result.

**Strengths:**

- Compared to previous one-stage methods, this method divides controllable generation into two steps.

**Weaknesses:**

- The motivation of this article is not clear. I hope the author can explain: 1) Why is the controllable generation divided into two steps? 2) Why use VQGAN for the generation model of the first step? 3) What are the advantages of the pretrained model? I see that the first step also requires loss optimization, which will also cause a training burden.
- The paper does not validate the effectiveness of the proposed method. 1) The final generated results do not align with the segmentation map, which makes me question the controllability of the method. 2) The method does not compare with ControlNet and T2I-adapter, which are state-of-the-art methods in controllable generation. 3) The authors did not choose indicators related to image quality in the experiments. 4) The authors did not verify the effectiveness of each part (including loss designs) of the designed framework separately.
- This paper is hard to read. The writing needs to be polished.

**Questions:**

Please see the weaknesses.

---

### Meta-Review · Area_Chair_3F1e · 2023-12-04

**Metareview:**

The final scores for this work are: Strong Reject $\times$ 3, Reject $\times$ 1. Overall, the opinions of the four reviewers are significantly skewed towards the negative. Additionally, most reviewers have pointed out serious shortcomings in the novelty of the work and the superiority and effectiveness of the proposed method. In other words, there is still substantial room for improvement in the completeness of the work, and further refinement in the writing may be necessary. Last but not least, the authors did not provide any rebuttals to the concerns mentioned above. Therefore, I decide to reject this work.

**Justification For Why Not Higher Score:**

Please refer to the metareview.

**Justification For Why Not Lower Score:**

N/A

---

### Decision · Program_Chairs · 2024-01-16

Reject